# Treatment of Actinic Keratosis: The Best Choice through an Observational Study

**DOI:** 10.3390/jcm11143953

**Published:** 2022-07-07

**Authors:** Seung-Ah Yoo, Yeong-Ho Kim, Ju-Hee Han, Chul-Hwan Bang, Young-Min Park, Ji-Hyun Lee

**Affiliations:** Department of Dermatology, Seoul St. Mary’s Hospital, College of Medicine, The Catholic University of Korea, Seoul 06591, Korea; useunga@hanmail.net (S.-A.Y.); catesal@naver.com (Y.-H.K.); alwaysmine8@gmail.com (J.-H.H.); mrbangga@catholic.ac.kr (C.-H.B.); 96015367@cmcnu.or.kr (Y.-M.P.)

**Keywords:** actinic, cryosurgery, imiquimod, keratosis, photochemotherapy

## Abstract

Actinic keratosis (AK) is a precancerous lesion that can progress to invasive squamous cell carcinoma if untreated. However, no gold standard treatment has been established. We aimed to investigate the management of AK by comparing the effectiveness and treatment duration of treatment modalities, including cryotherapy, imiquimod (IMQ), and photodynamic therapy (PDT). We reviewed the medical records of 316 patients diagnosed with AK at Seoul St. Mary’s Hospital from February 2015 to May 2020, and a total of 195 patients were included. The clearance rate was the highest in PDT, followed by cryotherapy and IMQ (82.4%, 71.2%, and 68.0%, respectively). The recurrence rate was the lowest in cryotherapy, followed by PDT and IMQ (3.5%, 6.7%, and 10.5%, respectively, *p* < 0.05). The average treatment duration was shortest with PDT, followed by IMQ and cryotherapy (5.5 weeks, 6.8 weeks, and 9.1 weeks, respectively, *p* < 0.05). The number of hospital visits was lowest for PDT, followed by cryotherapy and IMQ (1.8, 2.8, and 3.6, respectively, *p* < 0.05). PDT showed the highest clearance rate, a moderate recurrence rate, the shortest treatment duration, and the least number of visits, suggesting that PDT could be the first choice for treatment of AK. Considering the advantages as a topical agent, IMQ could also be a treatment option.

## 1. Introduction

Actinic keratosis (AK) is a precancerous lesion caused generally by exposure to ultraviolet radiation and has the potential to progress to squamous cell carcinoma (SCC) [1,2]. Studies comparing various treatments have been published; however, no gold standard treatment has been established to date [3,4,5,6].

The treatment modalities for AK can be broadly divided into lesion-targeted therapies and field-targeted therapies [2,4]. Lesion-targeted therapies, such as cryotherapy, ablative lasers, and surgical techniques, are usually mechanically destructive and suitable for single or smaller lesions [7]. Field-targeted therapy includes topical agents and procedural field therapies, such as photodynamic therapy (PDT), chemical peelings, and dermabrasion [8,9,10]. They can be applied to larger actinically damaged areas, including clinical and subclinical AK lesions, or to severe and numerous lesions that are burdensome to treat with lesion-targeted therapies.

Cryotherapy is the most commonly used lesion-targeted, destructive treatment, in which liquid nitrogen is applied to the lesion using a spray or cotton swab [2]. Cryotherapy is easy to implement and has the advantage of not having to rely on the patient’s compliance in the treatment process. Disadvantages include pain and the possibility of blistering, scarring, or hypopigmentation due to destruction of melanocytes.

Topical agents are the most commonly used field-targeted therapies, and several agents are available and approved by the Korean and the U.S. Food and Drug Administration (FDA) and European Medicines Agency (EMA) for treatment of AK, including 5% imiquimod cream (IMQ) [5,6,11,12]. IMQ is a Toll-like receptor 7 agonist, known to activate the immune system to produce cytokines [1]. When applied to skin lesion, IMQ rapidly increases the perilesional CD4+ T-helper with dendritic cells, CD8+ cytotoxic cells, CD68+ macrophages, and CD20+ B lymphocytes. Then, the cytotoxic T-cell-mediated immune response is likely to be activated by CD8+ T lymphocytes [13]. Local irritation reactions, such as pain, pruritus, and swelling, are possible adverse effects; rarely, systemic events, such as cardiovascular disorders, myalgia, arthralgia, and flu-like symptoms, may occur [14,15].

PDT is one of the approved procedural field therapies [5,6,10]. In PDT, when photosensitizer-treated skin is exposed to a light source of a suitable wavelength after several hours, the preferentially accumulated protoporphyrin IX in the more rapidly dividing atypical cells causes a phototoxic reaction and destroys the cells [10]. It is commonly used for facial AKs that occur in large areas because it has not only excellent cosmetic outcomes but also superior effectiveness. Disadvantages include erythema, edema, and a burning sensation during treatment [4,5,16].

In 2017, the British Association of Dermatologists published guidelines for the care of AK, recommending 5-fluorouracil, IMQ 5% cream, diclofenac gel, ingenol mebutate cream, cryosurgery, and PDT as strength of recommendation A [4]. In 2021, Eisen et al. published new guidelines for the management of AK, strongly recommending UV protection, topical IMQ, topical 5-fluorouracil, and cryosurgery, with conditional recommendation on PDT and diclofenac [5].

In this study, we investigated which method is more effective by comparing the effectiveness, recurrence rate, treatment periods, and the number of hospital visits for treatment of three treatment modalities that have been proven effective: cryotherapy, IMQ, and PDT [4,5,6].

## 2. Materials and Methods

### 2.1. Study Population and Enrollment Criteria

We reviewed the medical records of all patients diagnosed with AK at Seoul St. Mary’s Hospital from February 2015 to May 2020, which totaled 316 patients. Inclusion criteria were (1) patients diagnosed with AK clinically and histologically through skin biopsy, (2) patients treated with one of these treatment options: cryotherapy, IMQ, and PDT. Exclusion criteria were (1) patients treated with two or more modalities, (2) patients diagnosed with Bowen’s disease or SCC through re-biopsy, (3) patients with no more than 6 months of follow-up after the last treatment. A final total of 195 patients were included. All methods were carried out in accordance with relevant guidelines and regulations. This study was approved by the Institutional Review Board of Seoul St. Mary’s Hospital, the Catholic University of Korea (approval # KC19RESI0562).

### 2.2. Treatment Protocols

Cryotherapy: Liquid nitrogen was applied to the AK lesion with a spray, average duration 5–7 s, in three freeze–thaw cycles. The intervals between treatments were 4 weeks, and up to three sessions were performed until the lesions improved. The outcome assessment was performed by a dermatologist at week 12, followed by a follow-up up to 24 weeks after outcome assessment [17,18,19].

5% Imiquimod: The patients self-applied the cream once daily for 3 days per week, and the follow-up intervals were 2 to 4 weeks until the lesions improved. The outcome assessment was performed by a dermatologist at week 12, followed by a follow-up up to 24 weeks after outcome assessment [14,15,20].

Photodynamic therapy: Methyl aminolevulinate (MAL) 160 mg/g cream was applied to the treatment area, which was then covered with an occlusive dressing to prevent contact with (UV) light. After 3 h, the area was illuminated with Waldmann PDT 1200L (600–720 nm, 150 J/cm^2^). The first follow-up was 2 weeks after the first session. The second session was performed for patients whose lesions remained at the first follow-up. The outcome assessment was performed by a dermatologist 4 weeks after the last treatment, followed by a follow-up up to 24 weeks after outcome assessment [18,19].

### 2.3. Primary and Secondary Outcomes

The primary outcome of this study was treatment success at the final follow-up, which was assessed by a dermatologist and presented as ‘clearance rate’. It represents the proportion of patients with no gross evidence of residual lesions at outcome assessment and no recurrence during the subsequent 24 weeks of follow-up (Clearance rate = improved at outcome assessment and not recurred patients/total patients). The secondary outcomes of this study included recurrence rate, the number of visits, and the duration of treatment. The recurrence rate represents the proportion of patients whose lesions recurred during the follow-up period among the patients who improved at outcome assessment (Recurrence rate = recurrent patients/patients improved at outcome assessment). In terms of the number of visits, the visits for the treatment were included, and the visit for the outcome assessment was excluded.

### 2.4. Data Collection and Statistical Analysis

Characteristics of the enrolled patients (sex, age, location of the lesions, size and clinical grade of the lesions, treatment chosen), the date of each treatment and follow-up, clearance rates, recurrence rates, the average treatment duration, and the number of hospital visits for treatment were collected. We measured the size and grade of the lesion by reviewing clinical and dermoscopic photos of the enrolled patients; grade 1 describes slightly thick AK, grade 2 shows moderately thick AK, and grade 3 is very thick, hyperkeratotic and/or obvious (Figure 1) [21]. For statistical analysis, continuous variables, between-group differences were compared with the use of analysis of variance, and chi-square test and Fisher’s exact test were used to compare the clearance rate and the recurrence rate between the treatment groups with *p* values. Analyses were performed with the use of SPSS software, version 23.0 (IBM Corp., Armonk, NY, USA). In all statistical analyses, statistical significance was set at *p* < 0.05.

## 3. Results

### 3.1. General Characteristics of the Study Population

Table 1 shows the general characteristics of the study population for each treatment modality, and over half of patients (153/195, 78.4%) received cryotherapy. Most of the patients treated with PDT were female (98.8%), and the mean age of the group was over 5 years younger than that of the other groups, and all patients had lesions on their faces. There was a statistically significant difference in lesion size, with the PDT group showing the largest lesion size, and there was no difference between each group in the clinical grading of lesions.

### 3.2. Clearance Rate, Recurrence Rate, Average Treatment Duration, and Number of Visits for Each Treatment Modality

As shown in Table 2, PDT showed the highest clearance rate followed by cryotherapy and IMQ (76.5%, 71.2%, and 68.0%, respectively). However, there were no statistically significant differences. In terms of recurrence, cryotherapy had the lowest recurrence rate, followed by PDT and IMQ (3.5%, 7.1%, and 10.5%, respectively), and they were statistically significant. PDT was effective in 13 out of 17 patients, and 1 patient experienced recurrence. Cryotherapy was effective in 109 out of 153 patients, and 4 patients experienced recurrence. IMQ was effective in 17 out of 25 patients, and 2 patients experienced recurrence. The average treatment period was shortest for PDT, followed by IMQ, and cryotherapy (5.5 weeks, 6.8 weeks, and 9.1 weeks, respectively), and the number of visits for treatment was the lowest for PDT, followed by cryotherapy and IMQ (1.8, 2.8, and 3.6, respectively), and both were statistically significant.

## 4. Discussion

In this study, PDT showed the highest clearance rate, a moderate recurrence rate, the shortest treatment duration, and the lowest number of visits. Accordingly, PDT could be considered as the first-choice treatment out of the three modalities studied.

According to a randomized intra-individual controlled trial of Toqsverd-Bo et al., PDT is more effective than IMQ in AK clearance rate, with fewer local adverse effects [22]. On the other hand, in a small randomized intra-individual pilot study of Cortelazzi et al., MAL-PDT showed slightly lower clearance rate but lower recurrence rate than IMQ [23]. In this study, PDT was selected for patients who had multiple and/or large-sized facial AKs in consideration of cosmetic aspects. The composition of this patient group was expected to negatively affect the clearance rate. Nevertheless, PDT had the highest clearance rate with a moderate recurrence rate, suggesting that PDT could be the first choice for treatment of AK.

Although cryotherapy showed lower clearance rate than PDT in this study, it also could be a treatment option, considering the lowest recurrence rate with a relatively lower number of visits. 

For cryotherapy, Berman et al. compared the clearance rate of AK classified by the duration of spraying, number of freeze–thaw cycles, and distance from the tip of the spray to AKs [7]. Their results showed that higher clearance rate occurred with longer freeze times and use of a second cycle rather than a single cycle. The more aggressive the treatment, the higher the clearance rate. In the cryotherapy protocol of this study, we might be able to treat more aggressively to increase the clearance rate; however, most of our patients had lesions on their faces, so the possibility of scarring or hypopigmentation should have been considered. To increase the effectiveness of cryotherapy without being overly aggressive, several studies have suggested combining it with a topical agent, such as IMQ, or PDT [4,24,25,26]. It is supposed that the damaged stratum corneum due to cryotherapy facilitates the penetration of the topical agents or the photosensitizer, making it more effective.

In this study, IMQ seemed less preferable because of the lowest clearance rate and the highest recurrence rate. Our study revealed a clinical clearance rate of 68.0% in patients treated with IMQ. Jansen et al. reported that IMQ had treatment success in 75.8% of patients 3 months after the end of a treatment regimen in which IMQ was self-applied once daily, 3 days a week, for 4 consecutive weeks [3]. On the other hand, Tolley et al. reported the clearance rate of a 4-week course IMQ as 57.2% in their meta-analysis [23]. Although there were differences in the clearance rates, all three studies mentioned, including ours, treated in much the same way, and the common feature was that the clearance rate was lower than PDT. Nevertheless, IMQ showed relatively short treatment duration in this study and has the advantage of being convenient as a topical agent that is easy for patients to implement by themselves. Therefore, it could also be considered as a treatment option.

The objective of this study was to compare the therapeutic effectiveness and recurrence rate in clinical practice, as well as treatment duration and the number of visits, which might also have an impact on the preference or adherence of patients, subsequently on treatment choice. In addition to those stated in this study, there are more things to be considered, such as cost-effectiveness, patient preference for modality, or side effects. In other words, the selection of treatment method requires comprehension of these various aspects to obtain more satisfactory treatment process and results for both clinicians and patients. 

A limitation of this study is in the study design of the retrospective medical chart review, which resulted in the differences in sample size between the groups, groups with such small numbers, and non-randomized patients, leading to a possibility of selection bias [27].

However, a strength of this study lies in that, as far as we know, it is the first study that compared clearance rates, recurrence rates, treatment duration, and number of hospital visits of treatment modalities for AK in a single large institution. Therefore, the results, with a unified way of each treatment method, could be compared in various aspects. On the other hand, lesion-targeted therapy and field-targeted therapy were both included, and among field-targeted therapies, a topical agent and a procedural field therapy were both included as well. All three treatments compared were already proven to be highly effective in previous studies. To compare the exact effectiveness of each treatment, we only included patients who were treated with a single treatment modality.

Given the shortcomings of this study, further well-designed large-scale prospective studies comparing more various aspects would be helpful to set the gold standard treatment of AK.

## Figures and Tables

**Figure 1 jcm-11-03953-f001:**
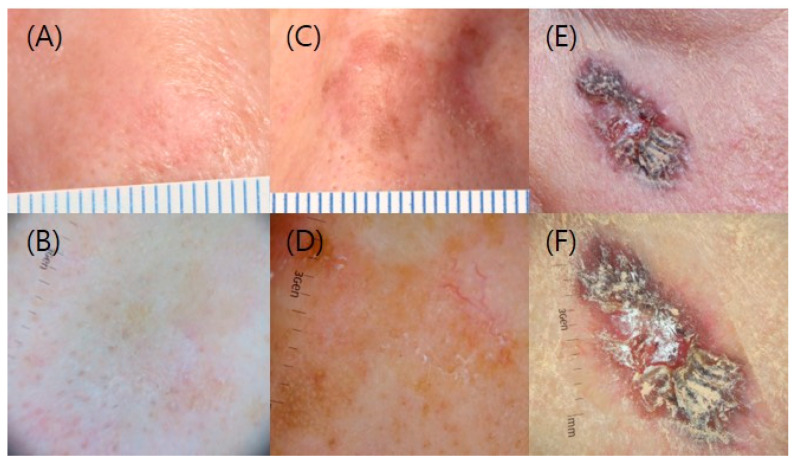
Clinical and dermoscopic grading of AK. (**A**,**B**) Grade 1; (**C**,**D**) Grade 2; and (**E**,**F**) Grade 3.

**Table 1 jcm-11-03953-t001:** General characteristics of the study population.

	Cryotherapy	IMQ	PDT	*p* Value
N	153	25	17	-
Male sex (%)	55 (36.0)	15 (60.0)	2 (1.2)	0.001
Age, years	71.1 ± 11.9	73.0 ± 11.7	64.4 ± 12.1	0.058
Location of lesions (%)				0.423
Scalp	1 (0.7)	1 (4.0)	0 (0.0)	
Face	142 (92.8)	22 (88.0)	17 (100.0)	
Body	10 (6.5)	2 (8.0)	0 (0.0)	
Lesion size (cm^2^)	2.7 ± 2.6	2.2 ± 1.8	4.3 ± 4.4	0.04
Clinical grade (1–3)	2.0 ± 0.8	1.9 ± 0.8	2.0 ± 0.8	0.399

Values are presented as number, mean ± standard deviation, or percentage. IMQ: imiquimod, PDT: photodynamic therapy.

**Table 2 jcm-11-03953-t002:** Clearance rates, recurrence rates, average treatment duration, and number of visits by treatment modality.

	Cryotherapy	IMQ	PDT	*p* Value
N	153	25	17	-
Clearance (%)	109 (71.2)	17 (68.0)	13 (76.5)	0.460
Recurrence (%)	4 (3.5)	2 (10.5)	1 (7.1)	0.031
Treatment duration, weeks (range)	9.1 ± 3.2 (4–12)	6.8 ± 3.7 (2–12)	5.5 ± 1.3 (2–6)	<0.001
Number of visits for treatment (range)	2.8 ± 0.4 (2–3)	3.6 ± 1.1 (2–5)	1.8 ± 0.4 (1–2)	<0.001

Values are presented as number or percentage. IMQ: imiquimod, PDT: photodynamic therapy.

## Data Availability

Data are contained within the article.

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
