# Peer review of "Treatment of Actinic Keratosis: The Best Choice through an Observational Study"

_jcm, 2022, doi:10.3390/jcm11143953_

Round 1

Reviewer 1 Report

This study covers only a limited area of Asia. Actinic keratosis (AK) is a common tumor worldwide, with different incidence rates among races, with Caucasians at higher risk. Caucasians with high rates of carcinogenesis from AK may consider surgical therapy.

Hence, ‘real-world evidenvce’ is an exaggeration.

Author Response

Thank you very much for your kind editorial letter of 05/30/2022.

We have attempted to carefully and thoroughly address all concerns raised by the editors and referees. With the help of your suggestions, we believe our manuscript has significantly improved.

Thank you very much for your consideration.

Sincerely,

Ji-Hyun Lee

# Reviewer 1

This study covers only a limited area of Asia. Actinic keratosis (AK) is a common tumor worldwide, with different incidence rates among races, with Caucasians at higher risk. Caucasians with high rates of carcinogenesis from AK may consider surgical therapy.

Hence, ‘real-world evidenvce’ is an exaggeration.

  • Response: Thank you for comment. As you requested, we have corrected the title.

Treatment of actinic keratosis: the best choice through an observational study

Reviewer 2 Report

This is a single center retrospective observational study comparing 3 types of treatment for AK: cryotherapy , IMQ and PDT. As also declared by the authors, this study have two significant limitations:

-the three groups have different sample sizes which have influence the statically significance 

- it is not clear if the AK treated were similar in all groups (size, grade)

I think that the power of the study should be improved, increasing the number of patients treated with IMQ and PDT to verify if the percentage of recurrence and clearance rate is always the same and significant. In this way, the result could be compared to the literature and useful to define PDT better than the others.

Author Response

Dear reviewer,

Thank you very much for your kind editorial letter of 05/30/2022.

We have attempted to carefully and thoroughly address all concerns raised by the editors and referees. With the help of your suggestions, we believe our manuscript has significantly improved.

Thank you very much for your consideration.

Sincerely,

Ji-Hyun Lee

Department of Dermatology, Seoul St. Mary’s Hospital, College of Medicine, The Catholic University of Korea

222, Banpo-daero, Seocho-gu, Seoul, 06591, Republic of Korea

yiji1@hanmail.net

# Reviewer 2

This is a single center retrospective observational study comparing 3 types of treatment for AK: cryotherapy , IMQ and PDT. As also declared by the authors, this study have two significant limitations:

-the three groups have different sample sizes which have influence the statically significance

- it is not clear if the AK treated were similar in all groups (size, grade)

I think that the power of the study should be improved, increasing the number of patients treated with IMQ and PDT to verify if the percentage of recurrence and clearance rate is always the same and significant. In this way, the result could be compared to the literature and useful to define PDT better than the others.

  • Response: Thank you for pointing this out. We fully agree that the difference between sample sizes would have affected statistical significance, so we mentioned this in discussion as a limitation of this study.

A limitation of this study is in the study design of retrospective medical chart review, which resulted in the differences in sample size between the groups, groups with so small numbers, and non-randomized patients, leading to possibility of selection bias

  • In addition, the lesion size and grade were measured for the similarity of AKs between groups and tested statistically for their differences. We added the results to Table 2.

Cryotherapy

IMQ

PDT

P value

N

153

25

17

-

Male sex (%)

55 (36.0)

15 (60.0)

2 (1.2)

0.001

Age, years

71.1 ± 11.9

73.0 ± 11.7

64.4 ± 12.1

0.058

Location of lesions (%)

0.423

Scalp

1 (0.7)

1 (4.0)

0 (0.0)

Face

142 (92.8)

22 (88.0)

17 (100.0)

Body

10 (6.5)

2 (8.0)

0 (0.0)

Lesion size (cm2)

2.7 ± 2.6

2.2 ± 1.8

4.3 ± 4.4

0.04

Clinical grade (1-3)

2.0 ± 0.8

1.9 ± 0.8

2.0 ± 0.8

0.399

Reviewer 3 Report

Reviewer recommendation: Major revision

Reviewer Comments:

Thank you for your retrospective review on the clearance and recurrence rates of three types of therapy for actinic keratosis. The manuscript is generally well written, but improvements can be made in the description of methods depth of discussion. Furthermore, the manuscript requires substantial stylistic changes prior to publication.

Please see specific comments & suggestions below:

Abstract:

1.     Please emphasize that the recurrence rate for PDT is statistically significant

2.     Please standardize the use of the oxford comma: “82.4%, 71.2% and 68.0% respectively”

3.     The sentences starting with “PDT” and “cryotherapy” are repetitive of information in the preceding sentences

4.     Suggest rephrasing or omitting “did not showed better results”

Introduction:

1.     Consider stylistic changes: “has the potential” rather than “known to have the potential”

2.     Remove “also” from second sentence.

3.     Consider stylistic changes in paragraph two: use parallel structure when describing the types of treatment modalities (i.e. examples of the modality, followed by clinical application)

4.     For clarity, consider rephrasing to “Disadvantages include pain and the possibility of blistering, scarring, or hypopigmentation due to …”

5.     Consider indicating that topical therapies are field-targeted therapies

6.     Combine the three sentences about IMQ into one paragraph, rather than having one paragraph of only one sentence

7.     Please briefly expand of the mechanism of action of IMQ

8.     Please expand on prior literature that has compared the effectiveness of therapeutic options

9.     Please edit the last sentence in the introduction to be more concise and clear. For instance, remove “tried to” and “details.”

Materials and Methods

10.  Remove “we confirm that”

11.  Please indicate the total follow up period for cryotherapy

12.  In the paragraph about IMQ, these sentences appear to be redundant: “…the effectiveness of the treatment was determined at 12 weeks” and “the outcome or treatment was assessed at 12 weeks”

13.  Please elaborate on how the treatment duration for IMQ was determined

14.  Please write out full name for MAL before using this acronym

15.  Consider stylistic changes: “which was then covered”

16.  Please use a superscript for the 2 in “cm2”

17.  PDT: for lesions that received an additional session of treatment, was there a follow up period after that, and how long? Did some patients receive more than two total sessions?

18.  Consider stylistic changes: “…determine treatment response and lesion recurrence”

19.  Please clarify this sentence: “the patients whose lesions recurred on the same sites which were on treatment during the follow up period”

20.  Suggest changing “not recurred” to “no recurrence” and removing “among total patients” as this is implied when you state “the proportion”

21.  Suggest changing to “…among the patients who improved at outcome assessment”  

22.  Please elaborate on why you have only measured duration of treatment for patients that improved.

Results

23.  Remove extra “.” after 3.1

24.  In the first sentence, consider using commas instead of two ands

25.  Consider stylistic changes: “over half of patients” rather than “the largest number of patients, over half of them”

26.  Change to “the proportion of female patients was…”

27.  Remove the extra “and” in the sentence beginning with “notably”. Remove the extra period in the next sentence.

28.  Rephrase “with no big differences”

29.  The mean number of visits for cryotherapy is 2, however in materials and methods you have written that outcomes are assessed after three sessions. Please clarify.

30.  In the table legend please indicate what the bracketed numbers in treatment duration represent. Also, under the PDT group 18 had lesions on the face, but there are only 17 people in this group, please address this. Consider including range in number of visits. Consider rounding to one significant digit rather than two.

31.  For clearance and recurrence rate paragraph, please use parallel sentence structures to report results on the three treatment modalities

Discussion

32.  General: The discussion should be better organized so that comparison with prior literature, explanation of findings, study limitations, and conclusions are grouped together. Please edit for grammar and repetition.

33.  Please indicate that while PDT showed better clearance rate this was not statistically significant

34.  Are the differences in number of visits and treatment duration statistically significant?

35.  Consider rephrasing to “Accordingly, PDT could be considered as the first choice treatment out of the three modalities studied”

36.  Please discuss whether the follow up period was equal for all modalities and whether follow up was of adequate length

37.  Consider citing other literature than the study by Gracia-Cazana et al., as neither DL PDT nor ingenol mebutate gel were studied in your work

38.  Please expand on why selecting PDT for multiple or large lesions considered its cosmetic advantages would negatively impact its effectiveness and provide a citation for this

39.  Please state the name of the organization who created the “2017 British guidelines”

40.  Some of the findings in discussion should be moved to the introduction. The statement that level of evidence for each treatment is 1+ or 1++ appears to contradict your statement that few studies have compared the effectiveness of therapeutic options for AK.

41.  The paragraph beginning with “Berman et al.” is interesting but does not appear to relate well to your study

42.  Suggest rephrasing sentence beginning with “and if topical…” for grammar, and providing a citation

43.  Please present results from table 3 in the results section of the paper. Also, please elaborate on your rationale for including these participants in the study if they met exclusion criteria

44.  Please shorten the paragraph beginning with “the objective of this study” and edit for grammar

45.  If you include Fitzpatrick skin type in the discussion, please report the Fitzpatrick skin type of the participants. Consider citing literature in this sentence on whether FST impacts the effectiveness of AK treatments

46.  In the discussion of limitations, please expand on how lack of randomization and lack of certain clinical information may have introduced bias

Author Response

Dear reviewer,

Thank you very much for your kind editorial letter of 06/01/2022.

We have attempted to carefully and thoroughly address all concerns raised by the editors and referees. With the help of your suggestions, we believe our manuscript has significantly improved.

Thank you very much for your consideration.

Sincerely,

Ji-Hyun Lee

Department of Dermatology, Seoul St. Mary’s Hospital, College of Medicine, The Catholic University of Korea

222, Banpo-daero, Seocho-gu, Seoul, 06591, Republic of Korea

yiji1@hanmail.net

# Reviewer 3

Thank you for your retrospective review on the clearance and recurrence rates of three types of therapy for actinic keratosis. The manuscript is generally well written, but improvements can be made in the description of methods depth of discussion. Furthermore, the manuscript requires substantial stylistic changes prior to publication.

Please see specific comments & suggestions below:

Abstract:

  1. Please emphasize that the recurrence rate for PDT is statistically significant
  2. Please standardize the use of the oxford comma: “82.4%, 71.2% and 68.0% respectively”
  3. The sentences starting with “PDT” and “cryotherapy” are repetitive of information in the preceding sentences
  4. Suggest rephrasing or omitting “did not showed better results”

  • Response: Thank you for careful review and specific comments. In order to emphasize statistical significance, we added p values. And we revised everything as you pointed out.

The recurrence rate was the lowest in cryotherapy, followed by PDT and IMQ (3.5%, 6.7%, and 10.5% respectively, P<0.05). The average treatment duration was shortest with PDT, followed by IMQ and cryotherapy (5.5 weeks, 6.8 weeks, and 9.1 weeks, respectively, P<0.05). The number of hospital visits was lowest for PDT, followed by cryotherapy and IMQ (1.8, 2.8, and 3.6 respectively, P<0.05). PDT showed the highest clearance rate, a moderate recurrence rate, the shortest treatment duration, and the least number of visit, suggesting that PDT could be the first choice for treatment of AK. Considering advantages as a topical agent, IMQ also could be a treatment option.

Introduction:

  1. Consider stylistic changes: “has the potential” rather than “known to have the potential”
  2. Remove “also” from second sentence.
  3. Consider stylistic changes in paragraph two: use parallel structure when describing the types of treatment modalities (i.e. examples of the modality, followed by clinical application)
  4. For clarity, consider rephrasing to “Disadvantages include pain and the possibility of blistering, scarring, or hypopigmentation due to …”
  5. Consider indicating that topical therapies are field-targeted therapies
  6. Combine the three sentences about IMQ into one paragraph, rather than having one paragraph of only one sentence
  7. Please briefly expand of the mechanism of action of IMQ
  8. Please expand on prior literature that has compared the effectiveness of therapeutic options
  9. Please edit the last sentence in the introduction to be more concise and clear. For instance, remove “tried to” and “details.”

  • Response: Thank you for careful review and specific comments. We revised everything as you pointed out.

Actinic keratosis (AK) is a precancerous lesion caused generally by exposure to ultra-violet radiation and has the potential to progress to squamous cell carcinoma (SCC).[1, 2] Studies comparing various treatments have been published, however, no gold standard treatment has been established to date.[3-6]

Treatment modalities for AK can be broadly divided into lesion-targeted therapies and field-targeted therapies.[2, 4] Lesion-targeted therapies such as cryotherapy, ablative lasers, and surgical techniques are usually mechanically destructive and suitable for sin-gle or smaller lesions.[7] Field-targeted therapy includes topical agents and procedural field therapies such as photodynamic therapy (PDT), chemical peelings, and dermabra-sion.[8-10] They can be applied to larger areas actinically damaged, including clinical and subclinical AK lesions, or to severe and numerous lesions that are burdensome to treat with lesion-targeted therapies.

Cryotherapy is the most commonly used lesion-targeted, destructive treatment, in which liquid nitrogen is applied to the lesion using a spray or cotton swab.[2] Cryothera-py is easy to implement and has the advantage of not having to rely on the patient’s com-pliance in the treatment process. Disadvantages include pain and the possibility of blis-tering, scarring, or hypopigmentation due to destruction of melanocytes.

Topical agents are the most commonly used field-targeted therapies, and several agents are available and approved by the Korean and the U.S. Food and Drug Administra-tion (FDA) and European Medicines Agency (EMA) for treatment of AK, including 5% imiquimod cream (IMQ).[5, 6, 11, 12] IMQ is a toll-like receptor 7 agonist, known to acti-vate the immune system to produce cytokines.[1] When applied to skin lesion, IMQ rap-idly increases the perilesional CD4+ T-helper with dendritic cells, CD8+ cytotoxic cells, CD68+ macrophages, and CD20+ B lymphocytes. Then the cytotoxic T-cell mediated im-mune response is likely to be activated by CD8+ T lymphocytes.[13] Local irritation reac-tions such as pain, pruritus, and swelling are possible adverse effects; rarely, systemic events like cardiovascular disorders, myalgia, arthralgia, and flu-like symptoms may oc-cur.[14, 15]

PDT is one of the approved procedural field therapies.[5, 6, 10] In PDT, when photo-sensitizer-treated skin is exposed to a light source of a suitable wavelength after several hours, the preferentially accumulated protoporphyrin IX in the more rapidly dividing atypical cells causes a phototoxic reaction and destroys the cells.[10] It is commonly used for facial AKs that occur in large areas because it has not only excellent cosmetic outcomes but also superior effectiveness. Disadvantages include erythema, edema, and a burning sensation during treatment.[4, 5, 16]

In 2017, British Association of Dermatologists published guidelines for the care of AK, recommending 5-fluorouracil, IMQ 5% cream, diclofenac gel, ingenol mebutate cream, cryosurgery, and PDT as strength of recommendation A.[4] In 2021, Eisen et al. published new guidelines for management of AK, strongly recommending UV protection, topical IMQ, topical 5-fluorouracil, and cryosurgery, with conditional recommendation on PDT and diclofenac.[5]

In this study, we investigated which method is more effective by comparing the effec-tiveness, recurrence rate, treatment periods, and the number of hospital visits for treatment of three treatment modalities that have been proven effective: cryotherapy, IMQ, and PDT.[4-6]

Materials and Methods

  1. Remove “we confirm that”
  2. Please indicate the total follow up period for cryotherapy
  3. In the paragraph about IMQ, these sentences appear to be redundant: “…the effectiveness of the treatment was determined at 12 weeks” and “the outcome or treatment was assessed at 12 weeks”
  4. Please elaborate on how the treatment duration for IMQ was determined
  5. Please write out full name for MAL before using this acronym
  6. Consider stylistic changes: “which was then covered”
  7. Please use a superscript for the 2 in “cm2”
  8. PDT: for lesions that received an additional session of treatment, was there a follow up period after that, and how long? Did some patients receive more than two total sessions?
  9. Consider stylistic changes: “…determine treatment response and lesion recurrence”
  10. Please clarify this sentence: “the patients whose lesions recurred on the same sites which were on treatment during the follow up period”
  11. Suggest changing “not recurred” to “no recurrence” and removing “among total patients” as this is implied when you state “the proportion”
  12. Suggest changing to “…among the patients who improved at outcome assessment”
  13. Please elaborate on why you have only measured duration of treatment for patients that improved.

  • Response: Response: Thank you for careful review and specific comments. We revised as much as possible to the things you pointed out.
    As for No. 17, after the second treatment, outcome assessment was done 4 weeks after the treatment without additional follow-up, and there was no more session after the second session. We mentioned it clearly and highlighted below.

2.1. Study population and enrollment criteria

We reviewed medical records of all patients diagnosed with AK at Seoul St. Mary’s Hospital from February 2015 to May 2020, which were total 316 patients. Inclusion criteria were 1) patients diagnosed with AK clinically and histologically through skin biopsy, 2) patients treated with one of these treatment options: cryotherapy, IMQ, and PDT. Exclu-sion criteria were 1) patients treated with two or more modalities, 2) patients diagnosed with Bowen’s disease or SCC through re-biopsy, 3) patients with no more than 6 months of follow-up after the last treatment. A final total of 195 patients were included. All meth-ods were carried out in accordance with relevant guidelines and regulations. This study was approved by the Institutional Review Board of Seoul St. Mary’s Hospital, the Catholic University of Korea (approval # KC19RESI0562).

2.2. Treatment protocols

Cryotherapy: Liquid nitrogen was applied to the AK lesion with a spray, average duration 5-7 seconds, in three freeze-thaw cycles. The intervals between treatments were 4 weeks, and up to three sessions were performed until the lesions improved. The outcome assess-ment was done by a dermatologist at week 12, followed by follow-up up to 24 weeks after outcome assessment.[17-19]

5% Imiquimod: The patients self-applied the cream once daily for 3 days per week, and the follow-up intervals were 2 to 4 weeks until the lesions improved. The outcome as-sessment was done by a dermatologist at week 12, followed by follow-up up to 24 weeks after outcome assessment.[14, 15, 20]

Photodynamic therapy: Methyl aminolevulinate (MAL) 160 mg/g cream was applied to the treatment area, which was then covered with an occlusive dressing to prevent contact with (UV) light. After 3 hours, the area was illuminated with Waldmann PDT 1200L (600-720 nm, 150 J/cm2). The first follow-up was 2 weeks after the first session. The second session was done for patients whose lesions remained at the first follow-up. The outcome assessment was done by a dermatologist 4 weeks after the last treatment, followed by fol-low-up up to 24 weeks after outcome assessment.[18, 19]

Results

  1. Remove extra “.” after 3.1
  2. In the first sentence, consider using commas instead of two ands
  3. Consider stylistic changes: “over half of patients” rather than “the largest number of patients, over half of them”
  4. Change to “the proportion of female patients was…”
  5. Remove the extra “and” in the sentence beginning with “notably”. Remove the extra period in the next sentence.
  6. Rephrase “with no big differences”
  7. The mean number of visits for cryotherapy is 2, however in materials and methods you have written that outcomes are assessed after three sessions. Please clarify.
  8. In the table legend please indicate what the bracketed numbers in treatment duration represent. Also, under the PDT group 18 had lesions on the face, but there are only 17 people in this group, please address this. Consider including range in number of visits. Consider rounding to one significant digit rather than two.
  9. For clearance and recurrence rate paragraph, please use parallel sentence structures to report results on the three treatment modalities.

  • Response: Thank you for careful review and specific comments. We revised as much as possible to the things you pointed out.
    As for No. 29, We specified in the method that up to three sessions were performed until the lesions improved.

3.1. General characteristics of the study population

Table 1 shows the general characteristics of the study population for each treatment modality, and over half of patients (153/195, 78.4%) received cryotherapy. Most of the patients treated with PDT were female (98.8%), and the mean age of the group was over 5 years younger than that of the other groups, and all patients had lesions on their faces. There was a statistically significant difference in lesion size, with the PDT group showing the largest lesion size, and there was no difference between each group in the clinical grading of lesions.

Table 1. General characteristics of the study population

Cryotherapy

IMQ

PDT

P value

N

153

25

17

-

Male sex (%)

55 (36.0)

15 (60.0)

2 (1.2)

0.001

Age, years

71.1 ± 11.9

73.0 ± 11.7

64.4 ± 12.1

0.058

Location of lesions (%)

0.423

Scalp

1 (0.7)

1 (4.0)

0 (0.0)

Face

142 (92.8)

22 (88.0)

17 (100.0)

Body

10 (6.5)

2 (8.0)

0 (0.0)

Lesion size (cm2)

2.7 ± 2.6

2.2 ± 1.8

4.3 ± 4.4

0.04

Clinical grade (1-3)

2.0 ± 0.8

1.9 ± 0.8

2.0 ± 0.8

0.399

3.2. Clearance rate, recurrence rate, average treatment duration, and number of visits for each treatment modality

As shown in Table 2, PDT showed the highest clearance rate followed by cryotherapy and IMQ (76.5%, 71.2%, and 68.0% respectively). However, there were no statistically significant differences. In terms of recurrence, cryotherapy had the lowest recurrence rate, followed by PDT and IMQ (3.5%, 7.1%, and 10.5% respectively), and they were statistically significant. PDT was effective in 13 out of 17 patients, and one patient experienced recurrence. Cryotherapy was effective in 109 out of 153 patients, and 4 patients experienced recurrence. IMQ was effective in 17 out of 25 patients, and 2 patients experienced recurrence. The average treatment period was shortest for PDT, followed by IMQ, and cryotherapy (5.5 weeks, 6.8 weeks, and 9.1 weeks, respectively), and the number of visits for treatment was the least for PDT, followed by cryotherapy and IMQ (1.8, 2.8, and 3.6, respectively), and both were statistically significant.

Table 2. Clearance rates, recurrence rates, average treatment duration, and number of visits by treatment modality.

Cryotherapy

IMQ

PDT

P value

N

153

25

17

-

Clearance (%)

109 (71.2)

17 (68.0)

13 (76.5)

0.460

Recurrence (%)

4 (3.5)

2 (10.5)

1 (7.1)

0.031

Treatment duration, weeks (range)

9.1 ± 3.2 (4-12)

6.8 ± 3.7 (2-12)

5.5 ± 1.3 (2-6)

< 0.001

Number of visits for treatment (range)

2.8 ± 0.4 (2-3)

3.6 ± 1.1 (2-5)

1.8 ± 0.4 (1-2)

< 0.001

Discussion

  1. General: The discussion should be better organized so that comparison with prior literature, explanation of findings, study limitations, and conclusions are grouped together. Please edit for grammar and repetition.
  2. Please indicate that while PDT showed better clearance rate this was not statistically significant
  3. Are the differences in number of visits and treatment duration statistically significant?
  4. Consider rephrasing to “Accordingly, PDT could be considered as the first choice treatment out of the three modalities studied”
  5. Please discuss whether the follow up period was equal for all modalities and whether follow up was of adequate length
  6. Consider citing other literature than the study by Gracia-Cazana et al., as neither DL PDT nor ingenol mebutate gel were studied in your work
  7. Please expand on why selecting PDT for multiple or large lesions considered its cosmetic advantages would negatively impact its effectiveness and provide a citation for this
  8. Please state the name of the organization who created the “2017 British guidelines”
  9. Some of the findings in discussion should be moved to the introduction. The statement that level of evidence for each treatment is 1+ or 1++ appears to contradict your statement that few studies have compared the effectiveness of therapeutic options for AK.
  10. The paragraph beginning with “Berman et al.” is interesting but does not appear to relate well to your study
  11. Suggest rephrasing sentence beginning with “and if topical…” for grammar, and providing a citation
  12. Please present results from table 3 in the results section of the paper. Also, please elaborate on your rationale for including these participants in the study if they met exclusion criteria
  13. Please shorten the paragraph beginning with “the objective of this study” and edit for grammar
  14. If you include Fitzpatrick skin type in the discussion, please report the Fitzpatrick skin type of the participants. Consider citing literature in this sentence on whether FST impacts the effectiveness of AK treatments
  15. In the discussion of limitations, please expand on how lack of randomization and lack of certain clinical information may have introduced bias

  • Response: Thank you for careful review and specific comments. We revised as much as possible to the things you pointed out.

    As for No. 40, we moved the part to the introduction and modified it.
    As for No. 41, because the clearance rate of cryotherapy in this study was found to be low, Berman's findings were mentioned as a way to improve it. And we also mentioned the limitations of aggressive cryotherapy, and then suggested the possibility of combination therapy as highlighted below.
    As for No. 43, we decided to exclude the results of table 3 from the paper, which was excluded by the exclusion criteria.
    As for No 45, because the patients' FST were not measured, we decided to delete that part.

In this study, PDT showed the highest clearance rate, a moderate recurrence rate, the shortest treatment duration, and the least number of visit. Accordingly, PDT could be con-sidered as the first choice treatment out of the three modalities studied.

According to a randomized intra-individual controlled trial of Toqsverd-Bo et al., PDT is more effective than IMQ in AK clearance rate, with fewer local adverse effects.[22] On the other hand, in a small randomized intra-individual pilot study of Cortelazzi et al., MAL-PDT showed slightly lower clearance rate but lower recurrence rate than IMQ.[23] In this study, PDT was selected for patients who had multiple and/or large-sized facial AKs in consideration of cosmetic aspects. The composition of this patient group was expected to negatively affect the clearance rate. Nevertheless, PDT had the highest clearance rate with a moderate recurrence rate, suggesting that PDT could be the first choice for treat-ment of AK.

Although cryotherapy showed lower clearance rate than PDT in this study, it also could be a treatment option considering the lowest recurrence rate with a relatively fewer number of visits.

For cryotherapy, Berman et al. compared the clearance rate of AK classified by the duration of spraying, number of freeze-thaw cycles, and distance from the tip of the spray to AKs.[7] Their results showed that higher clearance rate occurred with longer freeze times and use of a second cycle rather than a single cycle. The more aggressive the treat-ment, the higher the clearance rate. In the cryotherapy protocol of this study, we might be able to treat more aggressively to increase the clearance rate, however, most of our patients had lesions in their faces, so the possibility of scarring or hypopigmentation should have been considered. To increase effectiveness of cryotherapy without being overly aggressive, several studies have suggested combining it with a topical agent such as IMQ, or PDT.[4, 24-26] It is supposed that the damaged stratum corneum due to cryotherapy facilitates the penetration of the topical agents or the photosensitizer, making it more effective.

In this study, IMQ seemed less preferable because of the lowest clearance rate and the highest recurrence rate. Our study revealed a clinical clearance rate of 68.0% in patients treated with IMQ. Jansen et al. reported that IMQ had treatment success in 75.8% of pa-tients 3 months after the end of a treatment regimen in which IMQ was self-applied once daily, 3 days a week, for 4 consecutive weeks.[3] On the other hand, Tolley et al. reported the clearance rate of 4-week course IMQ as 57.2% in their meta-analysis.[23] Although there were differences in the clearance rates, all three studies mentioned including ours treated in much the same way, and the common feature was that the clearance rate was lower than PDT. Nevertheless, IMQ showed relatively short treatment duration in this study and has the advantage of being convenient as a topical agent that is easy for pa-tients to implement by themselves. Therefore it also could be considered as a treatment option.

The objective of this study was to compare the therapeutic effectiveness and recur-rence rate in clinical practice, as well as treatment duration and the number of visits, which also might have an impact on preference or adherence of patients, subsequently on treatment choice. In addition to those stated in this study, there are more things to be con-sidered such as cost-effectiveness, patient preference for modality, or side effects. In other words, selection of treatment method requires comprehension of these various aspects to obtain more satisfactory treatment process and results for both clinicians and patients.

A limitation of this study is in the study design of retrospective medical chart review, which resulted in the differences in sample size between the groups, groups with so small numbers, and non-randomized patients, leading to possibility of selection bias.[27]

However, a strength of this study lies in that, as far as we know, it is the first study that compared clearance rates, recurrence rates, treatment duration, and number of hos-pital visits of treatment modalities for AK in a single large institution. Therefore, results with a unified way of each treatment method could be compared in various aspects. On the other hand, lesion-targeted therapy and field-targeted therapy were both included, and among field-targeted therapies, a topical agent and a procedural field therapy were both included as well. All three treatments compared were already proven to be highly effective in previous studies. To compare the exact effectiveness of each treatment, we only includ-ed patients who were treated with a single treatment modality.

Given the shortcomings of this study, a further well-designed large-scale prospective studies comparing more various aspects would be helpful to set the gold standard treat-ment of AK.

Round 2

Reviewer 3 Report

The authors have done a good and comprehensive job of addressing reviewer comments. Accept.